# Diffusion Height and Order of Sulfur Dioxide and Bromine Monoxide Plumes from the Hunga Tonga–Hunga Ha'apai Volcanic Eruption

Qidi Li [1,2,†], Yuanyuan Qian [1,2,†], Yuhan Luo [1,*], Le Cao [3], Haijin Zhou [1], Taiping Yang [1], Fuqi Si [1] and Wenqing Liu [1]

1   Key Laboratory of Environmental Optics and Technology, Anhui Institute of Optics and Fine Mechanics, Hefei Institutes of Physical Science, Chinese Academy of Sciences, Hefei 230031, China
2   University of Science and Technology of China, Hefei 230026, China
3   Key Laboratory for Aerosol-Cloud-Precipitation of China Meteorological Administration, Nanjing University of Information Science and Technology, Nanjing 210044, China
*   Correspondence: yhluo@aiofm.ac.cn
†   These authors contributed equally to this work.

**Abstract:** A violent volcanic eruption attracting considerable attention occurred on 15 January 2022 near the South Pacific island nation of Tonga. To investigate its environmental impact, we retrieved the sulfur dioxide ($SO_2$) and bromine monoxide (BrO) vertical column densities of environmental trace gas monitoring instrument 2 (EMI-2) based on the differential optical absorption spectroscopy algorithm. The results showed westward and southeastward transport of principal parts of $SO_2$ and BrO plumes, respectively, from the Hunga Tonga–Hunga Ha'apai (HTHH) eruption. On 15 January, most of the released $SO_2$ entered the stratosphere (above 20 km) directly and spread rapidly westward (approximately 30 m/s). In contrast, the principal portion of the BrO spread southeastward slowly (approximately 10 m/s) within the 8–15 km altitude layer on 16 January. Our research results also suggest that during the HTHH eruption, BrO was released from the magmatic melt later than $SO_2$. The total $SO_2$ emissions from this eruption were approximately 0.24 Tg. The majority of $SO_2$ and BrO plumes were transported within the Southern Hemisphere. This study is an important extension to the empirical database of volcanological and magmatic degassing research.

**Keywords:** volcanic emissions; submarine eruptions; DOAS; EMI-2; sulfur dioxide plumes; bromine monoxide plumes

## 1. Introduction

Volcanic emissions have a significant influence on the local and global environment and climate. There was a decrease in greenhouse gases in the 2 years following the 1991 eruption of Mount Pinatubo, most likely due to changes in the radiative forcing of the stratospheric circulation [1], leading to a global average surface temperature drop of ~0.2–0.3 °C [2]. Due to sea-ice feedback, cooling can last for up to 16 years after a violent volcanic eruption in the Arctic region [3,4]. Volcanic activity also weakens the monsoon circulation [5–7] and inhibits the global water cycle [8,9]. A recent study suggested a strong link between the 2021 eruption of the La Soufriere volcano and the severe ozone depletion in the Antarctic in that year [10]. The La Soufriere volcanic eruption produced large amounts of stratospheric aerosols, which weakened the wave activity in the Southern Hemisphere stratosphere and reduced the Antarctic stratospheric temperature. They all influence the environment and climate through atmospheric dynamics and atmospheric chemistry. In short, the study of volcanic emissions is essential.

The Hunga Tonga–Hunga Ha'apai (HTHH) is a submarine volcano, and its eruptions destroyed the central part of the Island, and then the plume from the eruption rose into the

atmosphere. The plume from the HTHH volcanic eruption on 15 January 2022 extended over a maximum distance of 57 km vertically [11,12]. The plume is highly perturbing to the stratospheric aerosol and water vapor layers, generates large amounts of radiation, and potentially warms the surface [13,14]. The eruption produced an umbrella cloud similar to that of the 1991 eruption of the Pinatubo volcano and caused significant lightning and tsunamis [15–18], global surface seismic waves [19], and widespread atmospheric waves [17,20]. Investigations of volcanic emission compounds, such as sulfur dioxide ($SO_2$) and bromine monoxide (BrO), provide essential data on the volcanic eruption processes [21,22].

Relevantly, $SO_2$ not only affects human health (such as causing respiratory diseases and through acid rain irritating eyes and skin) but also induces climate and environmental change. The oxidation of $SO_2$ can lead to the formation of aerosols, increased haze, and photochemical smog [23]. The persistence of $SO_2$ varies from a few days to weeks depending on where it is in the atmosphere. $SO_2$ generally lasts for only a few days in the troposphere; however, in the lower stratosphere, it can last for several weeks or more [24]. $SO_2$ vertical column density (VCD) is an essential indicator of air quality and is closely linked to volcanic eruptions. The results of $SO_2$ VCD and emissions can provide data for air quality monitoring, a database for the traceability of $SO_2$ pollution and an early warning signal for volcanic eruption activity [25,26].

Bobrowski et al. first identified BrO in volcanic plumes [27] and found that bromine species can deplete ozone in the stratosphere and troposphere [28–30], which has a severe impact on the atmosphere and the climate [31,32]. Consequently, ground-based [33–37] and satellite [38,39] observations have been widely employed in studies on volcanic BrO emission. In addition, plume chemistry models have been developed to analyze the chemical composition of bromine from volcanic eruptions [40,41]. However, the process of bromine degassing is not yet adequately understood because of the paucity of observations and model data [21].

Methods for monitoring volcanic gases include airborne spectrometer measurements, chemical sensor measurements, and satellite observations [42]. As a remote sensing approach, satellite observations are the lowest risk and most labor efficient method as they do not require going in-person inside the active volcano area [22]. Since 1978, satellite remote sensing has contributed essential observations of volcanic eruption gases [43,44]. Carn et al. analyzed the contribution of volcanic $SO_2$ emission datasets to volcanology using satellite data [45]. Differential optical absorption spectroscopy (DOAS) was introduced by Platt et al. [46] during the 1970s and has been extensively used. It has been widely applied in ground-based [47], airborne, vehicle-borne [48], shipborne, and satellite-based [49] remote sensing. DOAS can measure trace gases such as ozone ($O_3$), BrO, formaldehyde (HCHO), $SO_2$, and nitrogen dioxide ($NO_2$) [50]. On 9 May 2018, China launched the Gaofen-5 (GF-5) satellite, which carried the country's first satellite-based environmental trace-gas monitoring instrument (EMI). The EMI measurements highlighted the validity and reliability of the instrument for retrieval of $SO_2$ VCD in a volcanic region [51]. EMI-2 is the second-generation model. In this study, we retrieved $SO_2$ and BrO VCDs from the EMI-2 load based on the DOAS algorithm.

This study analyzed the impact and spread of $SO_2$ and BrO from the HTHH volcanic eruption. We introduced the HTHH volcano and EMI-2, the retrieval methods and principles of $SO_2$ and BrO VCDs, and the calculation of $SO_2$ emissions. Then, we presented the $SO_2$ and BrO emission results from the HTHH eruption. In addition, we analyzed the wind field data to further investigate the distribution and transport processes of $SO_2$ and BrO. We also analyzed the sulfur and bromine emissions from HTHH volcano. Finally, we concluded that bromine was released from a magmatic melt later than sulfur during the eruption, and $SO_2$ and BrO had minor influence on the Northern Hemisphere.

## 2. Materials and Methods

### 2.1. The HTHH Volcano

The HTHH volcano (20.536°S, 175.382°W) is a submarine volcano located ~70 km northwest of the capital city of Tonga, Nukualofa. On 13 January 2022 at ~15:00 UTC (14

January 2022 at ~04:00 local time), the first eruption of the HTHH volcano sent a large plume of ashes and gases approximately 20 km above the stratosphere [52]. The HTHH volcano erupted strongly again at ~04:00 UTC (~17:00 local time) on 15 January 2022. The plumes from these eruptions extended over a maximum distance of 57 km, blanketing the surrounding land masses with ash and debris, and triggered severe tsunamis [53]. There was the third and smaller eruption at ~08:00 UTC (~21:00 local time) on 15 January 2022 [11].

### 2.2. EMI-2

EMI-2 is on board the GF-5 satellite, which was launched on 7 September 2021. EMI-2 is the highest spatial resolution remote sensing payload for atmospheric trace gases in China. EMI-2 utilizes four spectral channels: ultraviolet 1 (240–311 nm), ultraviolet 2 (311–401 nm), visible 1 (401–550 nm), and visible 2 (550–710 nm). In this study, the data for the retrieval of $SO_2$ were obtained from the ultraviolet-2 channel of EMI-2, with a data dimension of $1473 \times 211 \times 1072$ (time dimension × space dimension × spectral dimension). The spectral resolution of this channel ranges from 0.3 to 0.6 nm, with a nadir spatial resolution of $13 \times 24$ km$^2$. EMI-2 has a field of view of 114°, an altitude of 705 km in orbit, and a transit time of 10:30 am local time. Its descending node is at the equator.

### 2.3. Calculation of $SO_2$ and BrO VCDs

As stipulated by the Lambert–Beer law, when natural light penetrates the atmosphere, the solar spectrum changes and the concentration of trace gases in the atmosphere can be measured:

$$ln \frac{I^*(\lambda)}{I_0(\lambda)} = \sum [\sigma_j^*(\lambda) c_j L] = \sum [\sigma_j^*(\lambda) SCD_j] \tag{1}$$

where $I^*(\lambda)$ denotes the light intensity of the incident light passing through a gas layer of length $L$, $I_0(\lambda)$ represents the raw luminous intensity of the radiator, $\sigma_j^*(\lambda)$ represents the broadband absorption cross section at wavelength $\lambda$ of gas $j$, $c_j$ denotes the average concentration of gas $j$, $SCD_j = \int c_j L$ represents the slant column density (SCD) of $j$, and $D = \ln \frac{I^*(\lambda)}{I_0(\lambda)}$ denotes the differential optical density. The SCD of the desired trace gas can be obtained by least-squares fitting using Equation (1).

The $SO_2$ SCD was retrieved using the QDOAS system [46] in the band between 312 and 326 nm. The absorption cross-sections in this band include $SO_2$, $O_3$, $NO_2$, BrO, HCHO, and ring cross-sections. The ring cross-section was calculated using QDOAS software. Table 1 displays the detailed parameters for the $SO_2$ spectral fit. Qian et al. provided the $SO_2$ results from volcanic areas with EMI-2 [54]. Additionally, the $SO_2$ results were consistent with those from TROPOMI and they are both reliable.

**Table 1.** Detailed parameters for $SO_2$ spectral fits.

| Parameter | References |
|:---:|:---:|
| $SO_2$ | 298 K [55] |
| $O_3$ | 223 K, 243 K [56] |
| $NO_2$ | 220 K, 298 K [57] |
| BrO | 223 K [58] |
| HCHO | 298 K [59] |
| Ring | Calculated using QDOAS |
| Fitting Interval | 312–326 nm |
| Polynomial | 5 |

Additionally, we used a QDOAS system to retrieve the BrO VCD. Spectral fits of BrO were conducted in the 330–360 nm wavelength range, and the reference spectrum was the solar spectrum measured by EMI-2. The BrO, $O_3$, $NO_2$, $O_4$, and ring cross-sections were analyzed in the spectral fits. Table 2 lists the detailed parameters of the BrO spectral fits.

Figure 1 shows the spectral fits from the 50th dimension of a track (track number: 211027) on 16 January 2022. The differential SCD (dSCD) for BrO was $4.25 \times 10^{14}$ molec cm$^{-2}$, and the root mean square (RMS) for the residuals of the spectral fit was $1.26 \times 10^{-3}$.

**Table 2.** Detailed parameters for BrO spectral fits.

| Parameter | References |
|---|---|
| BrO | 223 K [58] |
| $O_3$ | 223 K, 243 K [56] |
| $NO_2$ | 298 K [57] |
| $O_4$ | 293 K [60] |
| HCHO | 298 K [59] |
| Ring | Calculated using QDOAS |
| Fitting Interval | 330–360 nm |
| Polynomial | 5 |

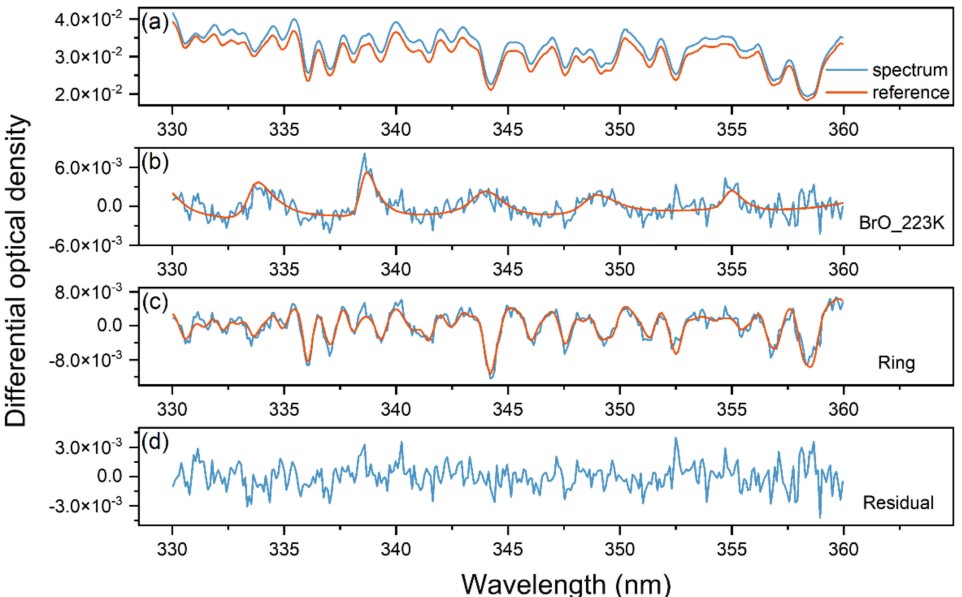

**Figure 1.** Spectrum fits of BrO on 16 January 2022. (**a**) measured (blue line) and reference spectrum (orange line); (**b**) measured (blue line) and fitted (orange line) BrO (223 K) optical density; (**c**) measured (blue line) and fitted (orange line) Ring optical density; and (**d**) the remaining residuals of the DOAS.

To obtain a conversion from SCD to VCD, we must consider the air mass factor (AMF), which is obtained by dividing SCD by VCD; that is, AMF = SCD/VCD. The lookup tables of the AMFs were obtained using the atmospheric radiative transfer model SCIATRAN. The input parameters for the look-up tables were: solar zenith angle (SZA), relative azimuth angle (RAA), viewing zenith angle (VZA), surface albedo, and cloud pressure. The surface albedo data considered in this research were obtained from a monthly mean climatology dataset based on OMI observations [61] with a spatial resolution of lat $\times$ lon = $0.5° \times 0.5°$. The AMF corresponding to the observed parameters of EMI-2 was obtained using multidimensional linear spatial interpolation, after which the VCD of sulfur dioxide was calculated.

### 2.4. Calculation of $SO_2$ Emissions

The emission flux [62] of the $SO_2$ plume from the volcanic eruption was obtained using the following equation:

$$F = \int_S c \cdot v \cdot n \cdot dS \qquad (2)$$

where F denotes the emission flux, c represents the concentration of $SO_2$ gas mass $(g/m^3)$, and v denotes the wind speed. In addition, n is a unit vector perpendicular to surface S.

The ERA5 dataset from the European Centre for Medium-Range Weather Forecasts website (https://cds.climate.copernicus.eu/, accessed on 17 November 2022) provided the wind and temperature fields data. Assuming a relatively stable wind field during the observation period, the emission flux of the $SO_2$ plume after it has moved in response to the wind field was obtained as follows:

$$F = \left( \sum_i V_{CD,i} \cdot l_i \right) \cdot v \cdot \cos(\theta) \tag{3}$$

where $l_i$ denotes the horizontal length of the *i*th pixel, and θ indicates the angle in the wind field direction along the longitudinal direction. For an $SO_2$ plume at a distance of *d* from the eruption point, the above equation can be rewritten as:

$$F = \left( \sum_i V_{CD,i} \cdot l_i \right) \cdot v \cdot \cos(\theta) \cdot \exp\left( \frac{d/v}{\tau} \right) \tag{4}$$

where $\tau$ denotes the duration of the $SO_2$ plume from the volcanic eruption. The emission of the $SO_2$ plume was obtained as follows:

$$M = F \cdot t \tag{5}$$

where M denotes the emission, and t represents the duration of the eruption.

### 3. Results

*3.1. Spread of $SO_2$ and BrO*

Figure 2 illustrates the different distributions and transports of $SO_2$ and BrO plumes from this eruption; panels (a), (c), (e), (g), (i), and (k) show the $SO_2$ VCD results from the retrieval of EMI-2 data for 14–19 January 2022.

On 14 January 2022, EMI-2 observations showed that $SO_2$ was mainly located in the 20°S–25°S, 170°W–180°W region, with a maximum value > 10 DU. On 15 January, the $SO_2$ mostly spread to the northwest, mainly located in the 15°S–25°S, 170°W–180°W, and 170°E–180°E region. The area with the highest $SO_2$ concentration inside the plume was located along several latitudes and longitudes around (23°S, 175°E), whereas the area with the lowest $SO_2$ concentration was between 170°W and 180°W. On 16 January, the principal part of $SO_2$ continued to spread westward and partially reached near 160°E. On 17 January, the $SO_2$ had reached near 140°E in northeastern Australia. The principal portion of $SO_2$ was located over the sea to the northeast of Australia, and the overall $SO_2$ above the ocean continued to spread northwestward. On 18 January, the principal part of $SO_2$ reached near 130°E, with a northwest–southeast distribution over the sea, while on 19 January, it reached approximately 110°E, having already crossed Australia. Overall, the spatial distribution of $SO_2$ VCDs from the EMI-2 retrieval for 14–19 January 2022 showed a westward transport of the principal part of the $SO_2$ plume from this eruption.

Panels (b), (d), (f), (h), (j), and (l) of Figure 2 show the distribution and transport of the BrO plume from this eruption for 14–19 January 2022. On 14–15 January, the BrO plume of HTHH submarine volcano was not significantly detected, possibly because there was a thimbleful of BrO emissions. On 16 January, the EMI-2 observations evidently indicated the two regions with BrO plume. Essentially, BrO from the HTHH volcanic eruptions was detected later than $SO_2$. The principal part of BrO was in the (17°S–30°S, 170°W–180°W) region, with maximum value above $1.5 \times 10^{14}$ molec $cm^{-2}$. The smaller part of BrO spread westwards into the region of (17°S–25°S, 170°E–178°E). On 17 January, the principal part of BrO spread to the southeast and reached near 40°S. In contrast, the smaller part of BrO continued to spread westwards and reached near 150°E. On 18 January, the principal part of BrO spread southeastwards to around 45°S, 160°W. The other part of BrO reached near

130°E in northeastern Australia. On 19 January, the principal part of BrO continued to spread eastwards and reached near 150°W, while the smaller part reached near 120°E in northwestern Australia. The BrO plume was gradually diluted, and the concentration range was mainly in $5.0 \times 10^{13}$–$10.0 \times 10^{13}$ molec cm$^{-2}$.

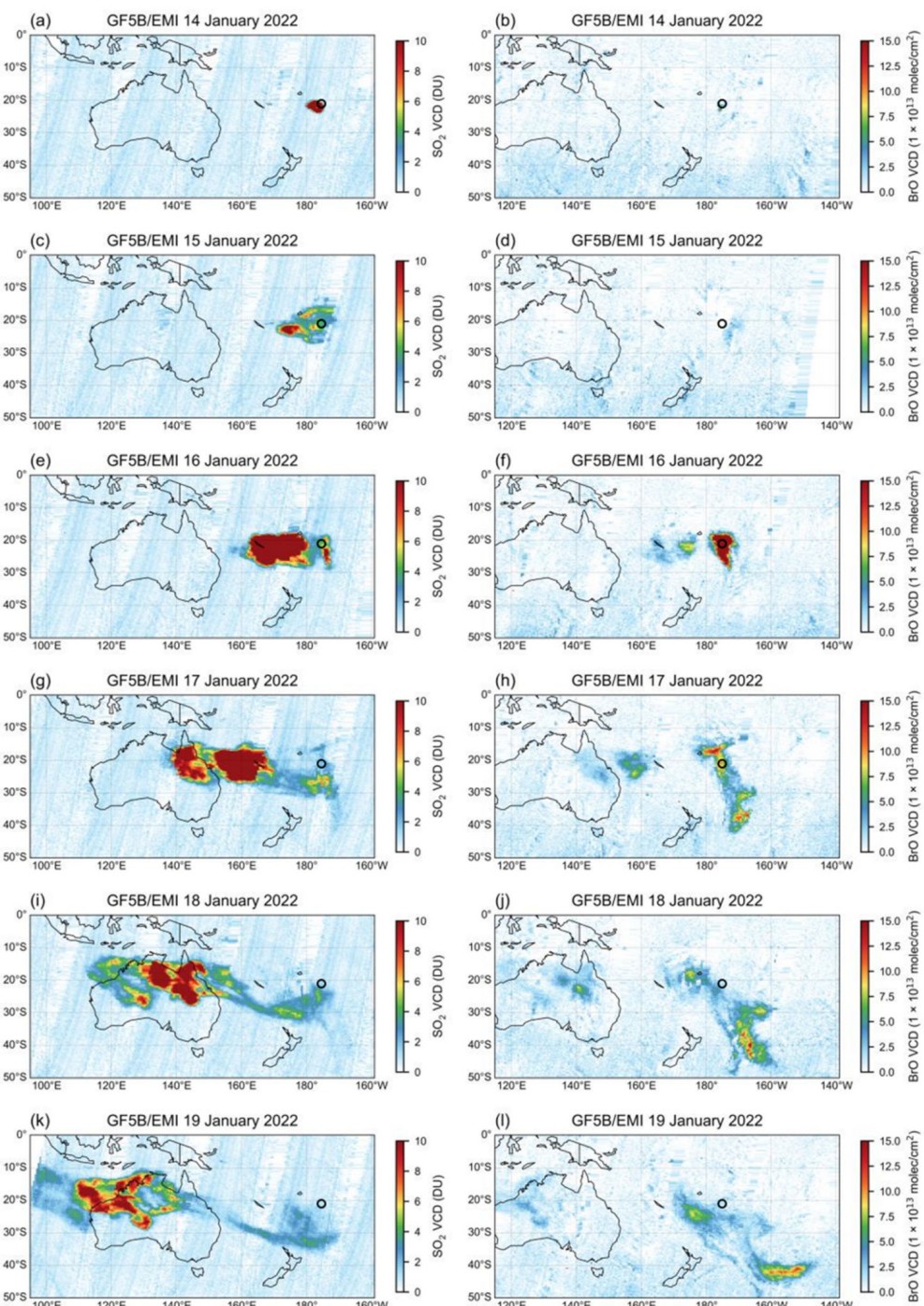

**Figure 2.** Monitored SO$_2$ and BrO VCDs and location (black circle) of the HTHH submarine volcano. The x label is the longitude and the y label is the latitude. (**a**) SO$_2$ VCD on 14 January 2022; (**b**) BrO VCD on 14 January 2022; (**c**) SO$_2$ VCD on 15 January 2022; (**d**) BrO VCD on 15 January 2022; (**e**) SO$_2$ VCD on 16 January 2022; (**f**) BrO VCD on 16 January 2022; (**g**) SO$_2$ VCD on 17 January 2022; (**h**) BrO VCD on 17 January 2022; (**i**) SO$_2$ VCD on 18 January 2022; (**j**) BrO VCD on 18 January 2022; (**k**) SO$_2$ VCD on 19 January 2022; (**l**) BrO VCD on 19 January 2022.

### 3.2. Relation of the Spread of SO₂ and BrO to Wind Fields

To further investigate the distribution and transport processes of $SO_2$ and BrO released from the HTHH submarine volcano eruptions, we analyzed the average vertical distribution of wind and temperature fields at an altitude of 0–32 km in the 20–25°S latitude zone on 14–19 January 2022. The results are presented in Figure 3. Appendix A shows the average vertical distribution of wind and temperature fields in the 15–20°S (Figure A1), 25–30°S (Figure A2), 30–35°S (Figure A3), and 35–40°S (Figure A4) latitude zones. The ERA5 dataset provided the wind and temperature field data used in this study.

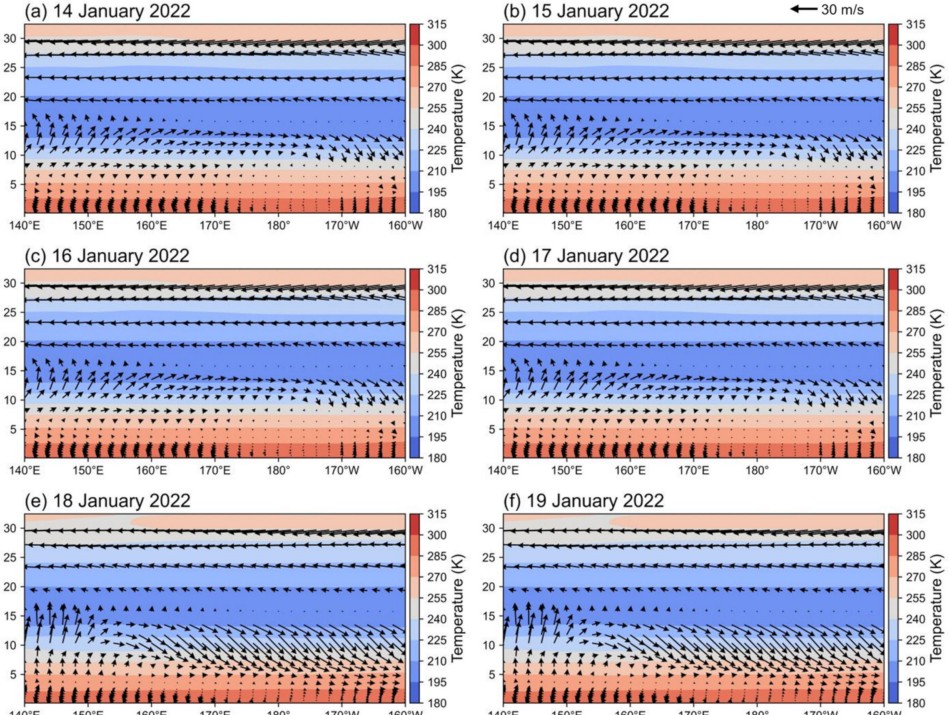

**Figure 3.** Average vertical wind field and temperature field at the altitude of 0–32 km in the 20–25°S latitude zone. The x label is the longitude and the y label is the latitude. (**a**) Vertical wind and temperature field on 14 January 2022; (**b**) vertical wind and temperature field on 15 January 2022; (**c**) vertical wind and temperature field on 16 January 2022; (**d**) vertical wind and temperature field on 17 January 2022; (**e**) vertical wind and temperature field on 18 January 2022; (**f**) vertical wind and temperature field on 19 January 2022.

As illustrated in Figure 3, on 15 January 2022, temperatures ranged from ~195–210 K at an altitude of 15–20 km, and wind speeds were low in the volcanic region. Above an altitude of 20 km, the wind in the atmosphere was predominantly easterly. In contrast, in the 8–15 km altitude range, the wind was predominantly westerly in the volcanic region. As wind speeds were very low below 8 km in the volcanic region, the diffusion of gases from the HTHH volcanic eruptions at this altitude was weak and, thus, was not considered in this study. In the volcanic region, $SO_2$ in the stratosphere gradually spread to the west. However, $SO_2$ in the troposphere spread to the southeast. Existing studies [44,63,64], of the HTHH eruption have identified a westward movement of the $SO_2$ plume. However, our study demonstrates that the $SO_2$ plume was moving southeastward in the troposphere.

The 15–20 km altitude layer showed different transmission dynamics. On 16 January, the relatively much higher concentration is caused by the upward motion and stronger convergence, which can bring more $SO_2$ and BrO into the upper layer. By contrast, there is downward motion east of 180°, which inhibits the vertical transport of $SO_2$ and BrO and results in a lot of deposition into the surface. Most of the $SO_2$ from the HTHH volcano eruptions spread significantly westward due to easterly winds and reached ~160°E. Thus,

the principal part of the $SO_2$ was above an altitude of 20 km. As wind speeds in the troposphere were significantly lower than those in the stratosphere, a small proportion of $SO_2$ at an altitude of 8–15 km spread slowly eastward and reached ~170°W. A small proportion of BrO was pushed above an altitude of 20 km and spread westward near 170°E. However, the principal part of the BrO was in the 8–15 km altitude range and spread southeastward to near 30°S, 170°W. On 17 January, stratospheric $SO_2$ continued to spread westward and reached close to 140°E. However, tropospheric $SO_2$ continued to spread southeastward and reached near 30°S. The tropospheric BrO spread consistently to the southeast, while the stratospheric BrO spread consistently to the west. On 18 January, the principal part of BrO spread southeastwards to around 160°W, and a small amount of $SO_2$ reached near 110°E. On 19 January, a large amount of $SO_2$ reached near 110°E, and the principal part of BrO continued to spread eastwards and reached near 150°W. Figure 4 shows the diagram of the three volcanic eruptions. We presume that $SO_2$ and BrO were erupted to different altitudes. This may be due to $SO_2$ and BrO being erupted with different intensities. As known from the references and satellite observations, a small amount of $SO_2$ and a thimbleful of BrO was erupted in the first eruption at ~15:00 UTC on 13 January (~04:00 local time on 14 January) 2022. Then, large amounts of $SO_2$ were violently erupted at ~04:00 UTC (~17:00 local time) on 15 January 2022, entering the stratosphere (up to 57 km) directly. Follow the westward wind above 20 km altitude, most $SO_2$ spread westward. However, the bulk of the BrO transmitted in the southeastward direction, which means there was little BrO during the second eruption. Thereby, most of the BrO was erupted in the third and weaker eruption at ~08:00 UTC (~21:00 local time) on 15 January 2022, where the eruption height was below 20 km. The southeastward wind field under 20 km provided reasonable explanation for the transmission of the BrO plume. However, the process of bromine degassing is not yet adequately understood.

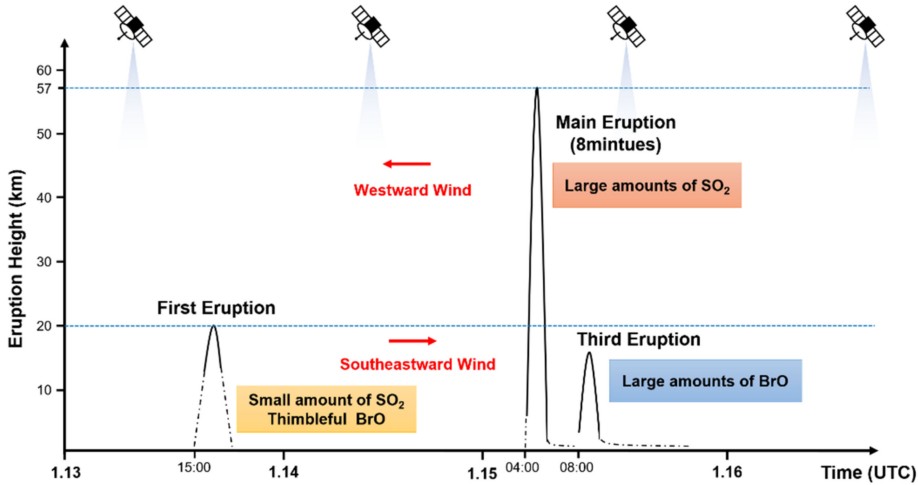

**Figure 4.** Diagram of the three volcanic eruptions.

## 4. Discussion

The violent volcanic eruption that occurred on 15 January 2022 near Tonga attracted considerable research attention, especially given that $SO_2$ and BrO levels and distribution patterns in volcanic emissions revealed essential information on the volcanic eruption processes. Here, we took advantage of the DOAS algorithm combined with EMI-2 data to investigate the impact and spread of $SO_2$ and BrO plumes from the January 15 volcanic eruption.

Notably, chlorine emissions have also been proposed as a crucial indicator for predicting volcanic eruptions [65–69]. Molten fluids of halogens (especially chlorine) and sulfur have also attracted a great deal of attention [70,71]. Chlorine had a lower melt-fluid partitioning rate than sulfur, concluding that chlorine was usually released from magma later than sulfur [72].

In contrast, the process of bromine degassing is inadequately understood; lack of data has stymied a clear consensus on whether bromine is released from the magma earlier than sulfur. BrO is not an important bromine species in magmatic gases, which are rapidly formed when the volcano plume enters the atmosphere. Due to the autocatalytic bromine explosion mechanism [73], the BrO concentration can be expected to increase quickly under sunlight through photolysis [74].

Bobrowski and Giuffrida [75] assumed that sulfur was released from magmatic melts after bromine. However, numerous studies have suggested that sulfur is released from magma melts earlier than chlorine. Later, Lübcke et al. [76] observed that volcanic bromine and sulfur emissions were associated with seismic events. They discussed two possibilities: bromine being released first or sulfur being released first, with each being equally plausible. Cadoux et al. [77] observed that the melt-fluid partitioning coefficients for bromine ranged from 3.8 to 20.8. This is comparable to that of chlorine, which has melt-fluid partition coefficients of 0.3–50 [78,79]. These values were both lower than the melt-fluid partition coefficients for sulfur, which ranged from 3 to 236 [80]. Thus, they suggested that bromine was consistent with chlorine and that bromine should be released after sulfur in magmatic melts. [77].

Our observations found that $SO_2$ was detected on 14 January, whereas most of the BrO was detected on 16 January. The BrO in the volcanic plume could be explained as rapidly produced by the autocatalytic inhomogeneous oxidation of HBr released from the volcanic eruption [32]. We concluded that bromine was released later than sulfur, from a magmatic melt during the eruption patterns of the HTHH volcano. Thus, further investigation of bromine could provide useful information for volcanologists.

As different transportation occurred in the 20–30 km and 8–15 km altitude layers, we analyzed average horizontal wind fields at these altitudes for the 14–19 January 2022 period (Figures 5 and 6). The Southern Hemisphere experiences summer during this period, with relatively straight easterly jet streams dominating the stratospheric wind field at low and medium latitudes. Most of the $SO_2$ released from the HTHH eruption entered the stratosphere directly and spread rapidly westward (approximately 30 m/s) via jet streams on 15 January. In contrast, only a small part of the BrO spread westward in this manner.

As shown in Figure 6, the tropospheric wind field was irregular, and wind speeds were significantly lower than those in the stratosphere. $SO_2$ in the troposphere spread slowly to the southeast between 14 and 19 January, with a transport interval mainly in the 10–40°S latitude band. Similarly, BrO in the troposphere spread slowly to the southeast because of northwesterly winds between 16 and 19 January, reaching near 40°S, 150°W. The principal portion of the BrO spread southeastward slowly (approximately 10 m/s) within the 8–15 km altitude layer on 16 January. Furthermore, because of the relatively independent wind circulation in the northern and southern hemispheres and the weak trans-equatorial transport, the majority of $SO_2$ and BrO was transported throughout the Southern Hemisphere.

As the $SO_2$ plume from this eruption had reached the stratosphere and had a long lifetime, Equation (4) can be replaced by Equation (3) to calculate the $SO_2$ emission from the HTHH volcanic eruption, where the wind field data are also from the ECMWF reanalysis dataset. The eruption on 15 January lasted 8 min and caused huge volumes of ash to rise rapidly and cover Tonga [81]. Using Equation (5), we estimated that the total $SO_2$ emissions from the 15 January eruption were ~0.24 Tg. In another study by Carn et al. [63], the initial estimate of the total $SO_2$ emissions released from this eruption on 15 January 2022 was ~0.4 Tg. However, our results may be lower because of subsequent small-scale eruptions. Compared to the 20 Tg $SO_2$ released by Mount Pinatubo (Philippines) in 1991 [82] and ~7.5 Tg $SO_2$ released by El Chichón in 1982 [83], the amount of $SO_2$ from the HTHH volcano was lower. The HTHH volcano has abundant brine that interacts with magma, unlike the Mount Pinatubo and El Chichón volcanoes. Therefore, a large amount of $SO_2$ released from the HTHH volcano was probably removed by wet deposition [84,85].

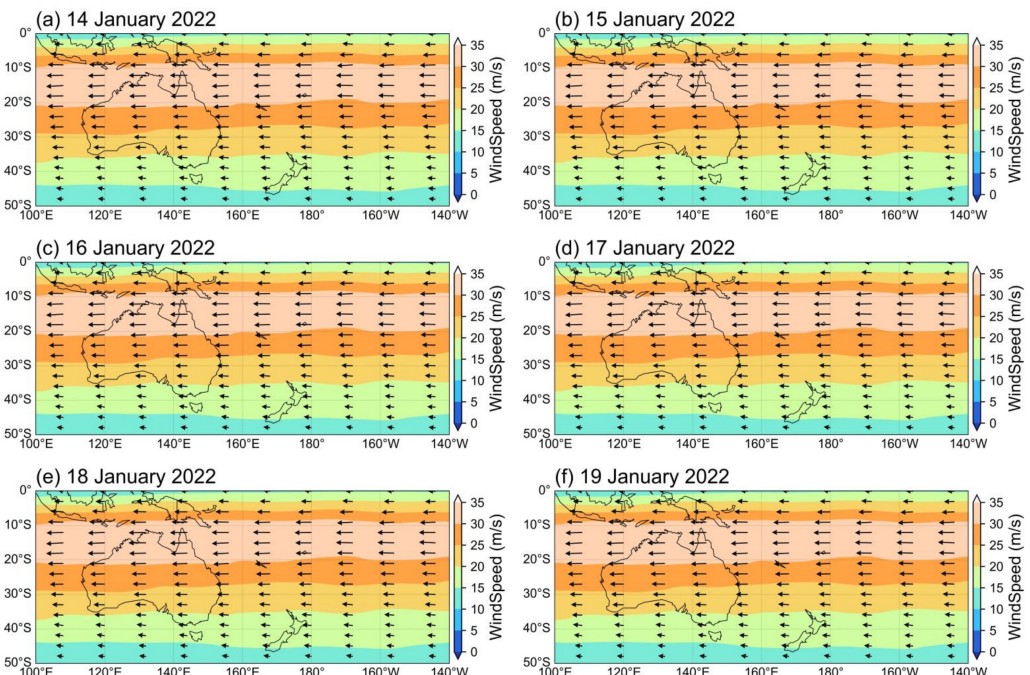

**Figure 5.** Average horizontal wind field at the altitude of 20–30 km. The x label is the longitude and the y label is the latitude. (**a**) Horizontal wind field on 14 January 2022; (**b**) horizontal wind field on 15 January 2022; (**c**) horizontal wind field on 16 January 2022; (**d**) horizontal wind field on 17 January 2022; (**e**) horizontal wind field on 18 January 2022; (**f**) horizontal wind field on 19 January 2022.

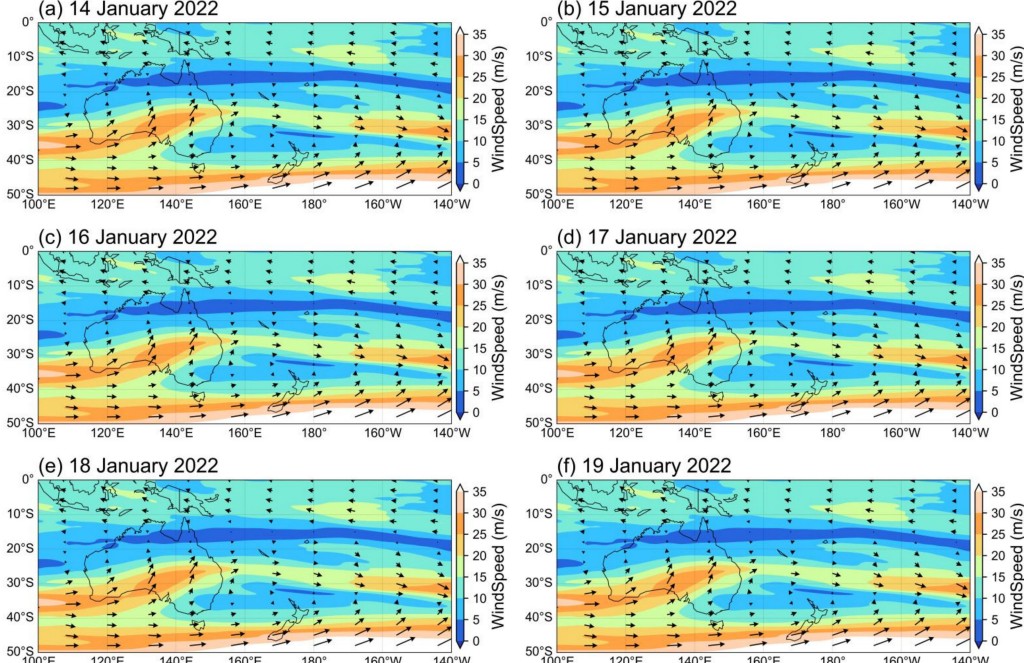

**Figure 6.** Average horizontal wind field at the altitude of 8–15 km. The x label is the longitude and the y label is the latitude. (**a**) Horizontal wind field on 14 January 2022; (**b**) horizontal wind field on 15 January 2022; (**c**) horizontal wind field on 16 January 2022; (**d**) horizontal wind field on 17 January 2022; (**e**) horizontal wind field on 18 January 2022; (**f**) horizontal wind field on 19 January 2022.

In summary, because of the relatively independent wind circulations of the Northern and Southern Hemispheres as well as the weak trans-equatorial transport, a major portion of $SO_2$ and BrO was transported within the Southern Hemisphere. Additionally, $SO_2$

emissions from the HTHH volcano were low, and thus the effects of $SO_2$ and BrO on the Northern Hemisphere were minor.

## 5. Conclusions

Using the DOAS algorithm, we successfully retrieved EMI-2 $SO_2$ VCDs and BrO VCDs for the HTHH volcanic eruption. Overall, the spatial distribution of $SO_2$ VCDs from the EMI-2 retrieval for 14–19 January 2022 showed westward transportation of the principal part of the $SO_2$ plume. On 16 January, EMI-2 observations indicated two regions where the BrO plume was apparent. The principal part of BrO spread slowly southeastward, whereas a small part of BrO spread rapidly westward.

Most of the $SO_2$ released from this eruption entered the stratosphere (above 20 km) directly and spread rapidly westward by jet streams. In contrast, the principal part of BrO was in the 8–15 km altitude layer and spread slowly southeastwards. Additionally, based on our observations, we conclude that bromine was released later than sulfur, from a magmatic melt during this eruption.

The lower $SO_2$ emission from the January 15 HTHH volcanic eruption, equal to ~0.24 Tg, compared to that of Mount Pinatubo (1991) and El Chichón (1982) eruptions, can be attributed to the fact that the HTHH volcano is submarine type with abundant seawater available for magma interaction. Moreover, a major part of $SO_2$ and BrO was transported in the Southern Hemisphere. Additionally, the influence of $SO_2$ and BrO on the Northern Hemisphere was found to be minor. This study is an important extension of an empirical database of volcanological and magmatic degassing research. In the future, integrated analyses based on numerical models and observational data will be necessary to study volcanic activity and its impact on climate change and the ecological environment.

**Author Contributions:** Methodology, Q.L., Y.Q. and Y.L.; Investigation, Q.L. and Y.Q.; Software, Q.L. and Y.Q.; Formal analysis, Q.L., Y.Q., Y.L. and L.C.; Validation, Y.L., L.C. and F.S.; Visualization, Q.L. and Y.Q.; Writing, Q.L. and Y.L.; Reviewing, Y.L.; Editing, Y.L.; Resource, Y.L., H.Z., F.S. and T.Y.; Funding acquisition, Y.L.; Supervision, Y.L., F.S. and W.L. All authors have read and agreed to the published version of the manuscript.

**Funding:** This study was financially supported by the National Natural Science Foundation of China (Grant Nos. 41941011 and 41676184) and the Youth Innovation Promotion Association of CAS (Grant No. 2020439).

**Data Availability Statement:** The data used in this research are available from Yuhan Luo from AIOFM, CAS (yhluo@aiofm.ac.cn).

**Acknowledgments:** We are thankful to the Ministry of Ecology and Environment Satellite Application Center for Ecology and Environment (SACEE) for providing the EMI-2 level 1 data. We gratefully thank the BIRA for providing the QDOAS software. We also gratefully thank ECMWF for providing the wind-field data.

**Conflicts of Interest:** The authors declare no conflict of interest.

## Appendix A

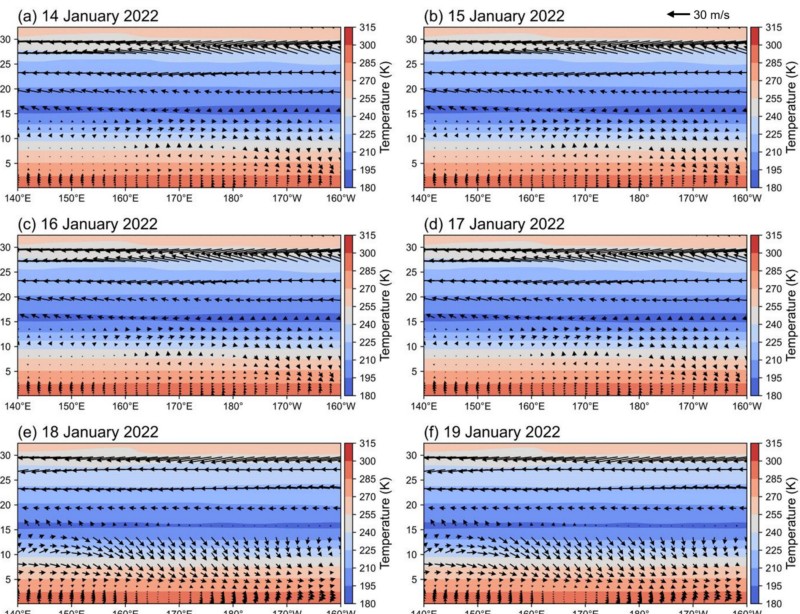

**Figure A1.** Average vertical wind field and temperature field at the altitude of 0–32 km in the 15–20°S latitude zone. The x label is the longitude and the y label is the latitude. (**a**) Vertical wind and temperature field on 14 January 2022; (**b**) vertical wind and temperature field on 15 January 2022; (**c**) vertical wind and temperature field on 16 January 2022; (**d**) vertical wind and temperature field on 17 January 2022; (**e**) vertical wind and temperature field on 18 January 2022; (**f**) vertical wind and temperature field on 19 January 2022.

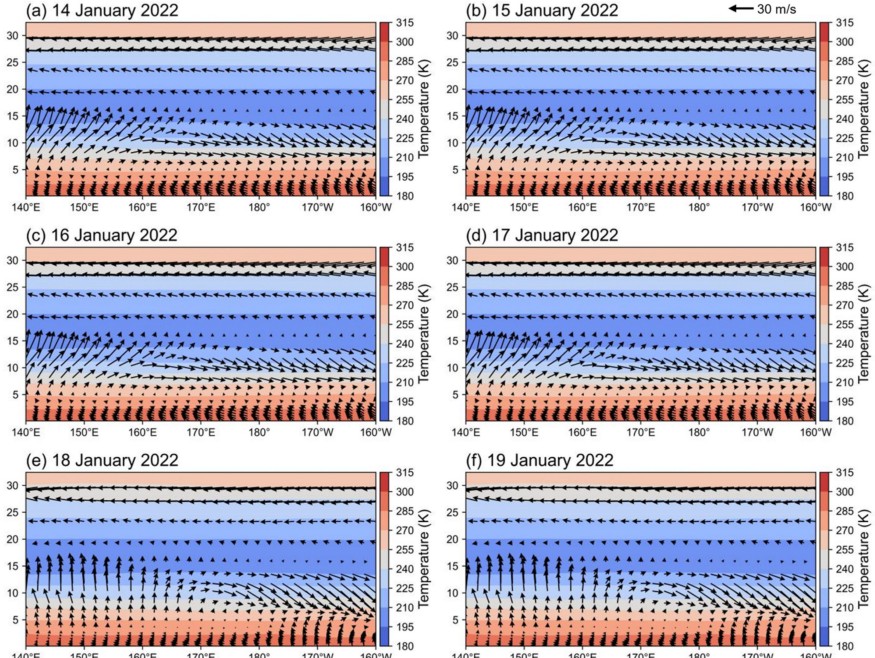

**Figure A2.** Average vertical wind field and temperature field at the altitude of 0–32 km in the 25–30°S latitude zone. The x label is the longitude and the y label is the latitude. (**a**) Vertical wind and temperature field on 14 January 2022; (**b**) vertical wind and temperature field on 15 January 2022; (**c**) vertical wind and temperature field on 16 January 2022; (**d**) vertical wind and temperature field on 17 January 2022; (**e**) vertical wind and temperature field on 18 January 2022; (**f**) vertical wind and temperature field on 19 January 2022.

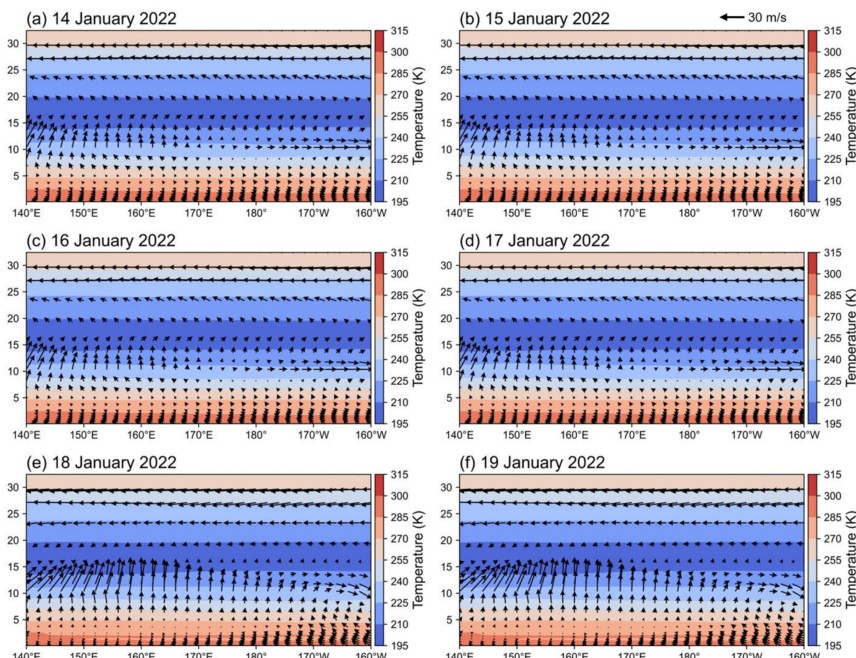

**Figure A3.** Average vertical wind field and temperature field at the altitude of 0–32 km in the 30–35°S latitude zone. The x label is the longitude and the y label is the latitude. (**a**) Vertical wind and temperature field on 14 January 2022; (**b**) vertical wind and temperature field on 15 January 2022; (**c**) vertical wind and temperature field on 16 January 2022; (**d**) vertical wind and temperature field on 17 January 2022; (**e**) vertical wind and temperature field on 18 January 2022; (**f**) vertical wind and temperature field on 19 January 2022.

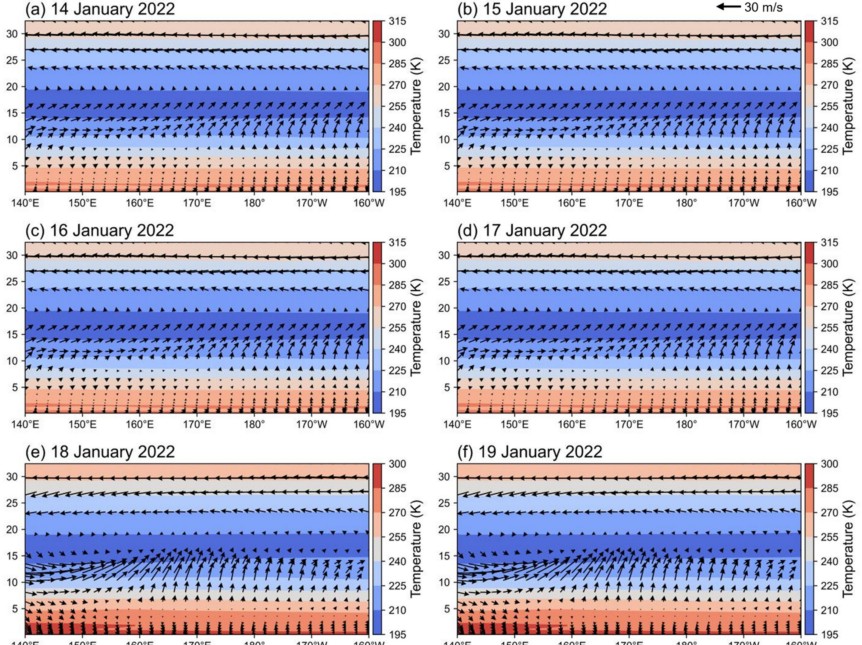

**Figure A4.** Average vertical wind field and temperature field at the altitude of 0–32 km in the 35–40°S latitude zone. The x label is the longitude and the y label is the latitude. (**a**) Vertical wind and temperature field on 14 January 2022; (**b**) vertical wind and temperature field on 15 January 2022; (**c**) vertical wind and temperature field on 16 January 2022; (**d**) vertical wind and temperature field on 17 January 2022; (**e**) vertical wind and temperature field on 18 January 2022; (**f**) vertical wind and temperature field on 19 January 2022.

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
