# Peer review of "Diffusion Height and Order of Sulfur Dioxide and Bromine Monoxide Plumes from the Hunga Tonga–Hunga Ha’apai Volcanic Eruption"

_remotesensing, doi:10.3390/rs15061534_

Round 1
Reviewer 1 Report
I read the paper of Li, Qian et al. with great interest as they analysed the concentrations of SO2 and BrO in the atmosphere following the big explosive eruption of the Hunga Tonga-Hunga Ha’apai volcano of last 15 January 2022. The authors presented their investigation of GaoFen-5 satellite with EMI-2 instrument to retrieve from spectral observations the concentration of SO2 and BrO, which is perfectly within the scope of the Journal. Furthermore, the paper adds some important data and discussion to this volcano eruption which is of great interest within the scientific community. The methods are well described, and the conclusions are supported by the results and compatible (for SO2) with independent estimations made with other satellites. So, for these reasons, I already recommend the paper to be published in Remote Sensing. On the other hand, I think there is space for some improvements and I provide below a list of some suggestions which I hope could be useful to the authors to improve and little expand their interesting work.
Major questions:
· Can you estimate the total amount of released BrO, please?
· About the Pinatubo volcano explosion and its consequences on climate and gas traces, I would suggest considering the following paper in your introduction and eventually compare with your result on Hunga Tonga-Hunga Ha’apai:
o Pitari, G.; Mancini, E. Short-Term Climatic Impact of the 1991 Volcanic Eruption of Mt. Pinatubo and Effects on Atmospheric Tracers. Nat. Hazards Earth Syst. Sci. 2002, 2, 91–108. https://doi.org/10.5194/nhess-2-91-2002
· Table 1. You need to describe better this table. 298K I guess is the equivalent black-body temperature with looks for SO2? What is ring.exe? It looks like an executable... you made a fit with a polynomial of 5 degrees, good, but you need to describe how you process the data according to these parameters and describe what the parameters are in the text or table caption (at your choice).
· It would be interesting if the authors could also investigate the vertical distributions of SO2 and BrO or at list the altitude of maximum concentration (you discussed something in lines 241-245, but I don’t understand how you get this information from literature or your data?). In the literature, this has been done for other volcano eruptions with multi-spectra satellite data similar to the one used by the authors of this paper (see, for example, Corradini et al., 2010). Anyway, as the instruments are different, I don’t know if this is possible with EMI-2, but if it’s possible it’s important information to understand better the big Hunga Tonga-Hunga Ha’apai volcano eruption of January 2022.
o Corradini, S., Merucci, L., Prata, A. J., and Piscini, A. (2010), Volcanic ash and SO2 in the 2008 Kasatochi eruption: Retrievals comparison from different IR satellite sensors, J. Geophys. Res., 115, D00L21, doi:10.1029/2009JD013634
· Did you investigated some days before the eruption searching for emission of Chlorine as you said that is a known volcano eruption precursor in lines 253-257. It would be interesting if you can add it.
· Can you provide in the discussion or conclusions a comparison with other satellite measurements of SO2 of this eruption. I know ESA provided some maps calculated with TROPOMI instrument onboard Sentinel 5-p. I think they further support your conclusions.
o https://www.esa.int/ESA_Multimedia/Images/2022/01/Sulphur_dioxide_from_Tonga_eruption_spreads_over_Australia
o At the link, you can find the picture on 18 January 2022 and a short description. If you compare the online picture of ESA with your Figure 2 (i) there is a great agreement. This is an independent measurement by another satellite, so it’s very good the results show the same distribution of SO2, which means that your results (and ESA’s) are both reliable!
Specific (minor) points:
· Line 18. I think the word “load” needs to be deleted or I don’t understand well the sentence.
· Line 21. Please invert “SO2 released” in “released SO2”
· Lines 22 and 23. I suggest specifying in brackets how much rapidly and slowly the plums propagate westward or southward.
· Line 38. Can you spend some more words on this very interesting reference [9]? Which mechanism do the authors suggest for the impact on the ozone hole from the volcanic activity?
· Line 39. The ocean surface is composed of water, how can be broken? Maybe the authors would say that it changed the emersed part of the Island (which is part of the top of the Hunga Tonga-Hunga Ha’apai volcanic cone), destroying the central part of the Island?
· Line 43 / Reference 11. I think the preprint you cited in reference 11 is finally published in this paper:
o Gupta, A.K., Bennartz, R., Fauria, K.E. et al. Eruption chronology of the December 2021 to January 2022 Hunga Tonga-Hunga Ha’apai eruption sequence. Commun Earth Environ 3, 314 (2022). https://doi.org/10.1038/s43247-022-00606-3
If so, please update the reference to the peer-reviewed paper. In any case, references of preprints are also okay, but if there is a peer-review published version, it’s preferable to cite this one.
· Line 46. “Acid rain” is not a “human disease”, but I think it could irritate eyes and skin, so rewrite better this sentence, please.
· Line 47. I suggest replacing “fuels” with “induces”
· Line 49. In English is more proper the term “persistence” than “duration” in this context.
· Line 58. Maybe the authors can add “and the climate” after “on the atmosphere”?
· Line 59. I suggest replacing “retrieval” with “emission”
· Line 67. Maybe the authors can add “as they don’t require going in person inside the active volcano area”.
· Line 90. If you refer to the main eruption, it was at 4 am UT of 15 January 2022 or 16 (4pm) of 14 January local time... (source: https://earthquake.usgs.gov/earthquakes/eventpage/us7000gc8r/executive). Please check, and I suggest using the UT also because the region is close to the date/change line... so it’s quite ambiguous in local time.
· Line 105. You need to add if it’s for ascending or descending node at the equator (I guess)
· Line 120. This sentence is not clear. Do you mean that Qian et al. provided the results from volcanic areas with EMI-2 or the calibration parameter to retrieve the SO2 from EMI-2 for volcano studies?
· Figure 2. I would like to suggest you to show the location of the volcano in all subpanels. To avoid covering the important data, you can use only the border of the symbol (If you use Matlab, you can set a MarkerEdgeColor, ‘k’ and MarkerFaceColor, ‘none’). This is just a suggestion, if you find that the graphic result is not good also, the present version of the figure is fine for me. Note: If you edit the figure 2, I suggest you also change the captions of BrO colorbar according to the Remote Sensing style, so 1e13 needs to be written as 1013 and cm^2 as cm2. I suggest you do this to avoid editing two times the same figure in final proofreading to have smoother processing of your paper, but for me, there is no problem also with the notation you used.
· Line 174. I suggest adding “inside the plume” after “SO2 concentration”
· Line 186. I think the authors mean “BrO emissions”, not “BrO eruptions”.
· Line 208. I suggest you to write “the vertical distribution of wind and temperature fields”.
· Line 210. Please add a sentence explaining how you considered the latitudinal spread from 10° S to 40° S. Did you calculate an average?
· Figure 3. You must add a reference arrow of the wind speed to understand how many m/s correspond the shown arrows. One for all the subgraph would be more than sufficient, but please add.
· Line 276. I suggest to write this sentence with some caution as this is the interpretation of the author according to the cite literature. So, for example: “The BrO in the volcanic plume could be explained as rapidly produced...”
· Line 283. I suggest adding a reference also to Figure 5, as in the text, you present the results of wind speed at the two altitudes that you showed in Figure 4 and Figure 5.
· Line 324. I suggest writing “was probably removed”, as you guessed, the interaction with water/brine removed it. I agree with your discussion, and it’s reliable, but as you don’t measure this process you need some caution writing.
· Line 342. I don’t think “at” is right in English, I suggest you to substitute it with “equal to”
Reviewer 2 Report
This study investigated the emission and transport of SO2 and BrO from the HTHH using satellite observations. The authors show a contrast of transport between SO2 and BrO: SO2 is mostly transported westward while BrO tends to spread southeastward. They argue that this is mainly because SO2 plumes enter the stratosphere and its spread is more modulated by the easterly wind while BrO is more located between 8-15 km under the control of the westerly wind. The results are interesting to me, and the manuscript is generally well-written. My major question is that the author should provide more evidence on why SO2 can enter the stratosphere directly while BrO stays much lower. I suggest this manuscript can be published in Remote Sensing after addressing this issue and some detailed comments shown below.
Specific comments:
P2L95: inappropriate citation in #43. I don’t think this cited paper has discussed any global climate impact due to the HTHH volcano.
P6L209: January->between January. Any reason why the authors just show the distribution on the first three days? I think all these six-day results need to be shown to readers as well as to keep consistency with other figures.
P8L219–221: The author repeats the sentence of “low wind speed below 8 km leads to weak diffusion”.
P8L223-224: How could the author get a sense of southerly/northerly wind from a figure averaged over latitude (Fig. 3). Please polish the writing here.
P8L228: The layer between 15–20km is in a transition zone between easterly and westerly for all three days. It is not unique on Jan 16. To me, the relatively much higher concentration on Jan 16 is because the upward motion and associated convergence are much stronger, which can bring more SO2/BrO into the atmosphere. By contrast, there is downward motions east of 180°, which inhibits the vertical transport of SO2/BrO and results in a lot of deposition into the surface.
P8L232–236: I suggest the author have a figure of the profile of SO2 and BrO. Are most of the concentrations located in the stratosphere rather than between 8–15 km?
P8L241–245: Why SO2 enters the stratosphere but not BrO? Is Figure 2 an indicator of emission or concentration intensity? The author should be more cautious about the terminology.
P8L247–249: I suggest the author show the profile of the SO2 and BrO.
P9L285–287: Again, why SO2 can enter Stratosphere directly but not for BrO.
Reviewer 3 Report
Review report on ‘Unexpected Spread of Sulfur Dioxide and Bromine Monoxide Plumes from the Hunga Tonga–Hunga Hapa’ai Volcanic Eruption’ by Qidi Li et al.
This manuscript presents the early evolution of the Sulfur Dioxide and Bromine Monoxide from the January 2022 Hunga-Tonga eruption based on environmental trace gas monitoring instrument 2 (EMI-2) measurements. The eruption was unique and got immense interest from the scientific community around the world. The topic of this study is interesting and worth to be published. The manuscript could be considered to be published in Remote Sensing after the following revision.
Why only 14-19 January? Authors can extend their analysis for at least 10 days after the major eruption.
Line 18: ‘environmental trace gas monitoring instrument 2 (EMI-2) load’? what do you mean by load? I don’t understand ‘load’ this.
Line 41: ‘extended over a maximum distance of 57 km’-change to ‘extended over a maximum distance of 57 km vertically’.
Lines 90-91: ‘On January 14, 2022, at ~ 4:00 am local time, the first eruption of the HTHH volcano sent a large plume of ashes and gases above the stratosphere.’ I don’t think it send ash and gases into the stratosphere on January 14. Please check it once. I would like to see the time in UTC.
Line 210: ‘The ECMWF dataset’? Which reanalysis? Is it ERA-5 or another reanalysis? Please mention it clearly.
In the Material and Methods section, authors need to write about winds and temperature data information.
Line 339: ‘spread lowly southeastwards’. Should be ‘slowly’. Correct it.
In Figure 2, the x and y labels are missing. Similarly, for Figures 4 and 5 also.
Reference no. 11 has already been published in final form as a publication. Please update it.
Round 2
Reviewer 1 Report
Dear authors,
I thank you so much for your careful reply to all my questions and for the new version of the manuscript, which I found improved with respect to the first submission. In particular, I appreciate so much the new Figure 4, which graphically illustrates well the three different eruptive episodes, altitudes reached by the plumes, substances in the plumes and time of the eruptions. I would like to ask the authors to include the following answer also in the main manuscript:
· Line 313. You can add your answer to my question (I adjusted some spelling) “From the Aura Ozone Monitoring Instrument (OMI) OClO product, in the case of Hunga Tonga-Hunga Ha’apai eruption, there were no unusual areas of OClO concentrations before and during the eruption (https://disc.gsfc.nasa.gov/datasets/OMOCLO_003/summary?keywords=OCLO, last access XX/XX/XXXX). This may be because this is a submarine volcano, and the target gas could have dissolved in the sea.”
For the rest of the manuscript, I think now it’s okay apart from some very minor points listed below that I would like the author could check, please (mainly of language). So globally I would suggest publishing the paper in Remote Sensing as a precious contribution from satellite observations of SO2 and BrO to the understanding of this last great Hunga Tonga-Hunga Ha’apai eruption that occurred in January 2022.
Specific (very minor) points:
· Lines 41 and 43. I am not sure, but I think it’s better you delete “The” in front of “La Soufriere”. This is because in French, “La” means “the”, so it sounds strange for me, but you can ask to a native English speaker/teacher how to deal with this case.
· Line 43. I think It’s better to say “in the same year” instead of “in that year”
· Line 45. I would suggest saying “These phenomena” instead of “They all”
· Line 124. Please change “It’s” with “The local time is”
· Line 262. I suggest replacing “is caused” with “was likely caused”
· Line 281. I think you can delete “had” using the same tense as the previous sentence.
· Line 285. “We assume” could be better than “we presume”
· Line 286. In addition to the intensity of the eruption, could the atomic weight of the two molecules (BrO and SO2) play a role in the altitude reached in the atmosphere? I made a calculation of atomic weight and noticed that BrO is heavier than SO2, so maybe this factor could also contribute to explaining the different altitudes, even though the main reason I agree with the author is related to the intensity of the eruption. If you agree, you can add a sentence about this.
Br 79.904 Dalton, O: 15.999 Dalton, S: 32.06 Dalton. BrO: 95.903 Dalton, SO2: 64.058 Dalton
· Line 295 “provides” (with s third person)
· Appendix A. I thank you for the addition of this section that contributes to improving the quality of the paper. I just have a doubt that as an appendix, I expect you to insert some text to comment on these figures (also within the appendix, not only in the text). Anyway, I leave it to the editor and authors the decision how to consider this part, or better what do among the following three options: 1) Leave as it is now (Appendix with no text); 2) Leave as an appendix and add some short description of the figures; 3) Leave as it is but shift in Supplementary Materials.